# Genome analysis of *Legionella pneumophila* ST23 from various countries reveals highly similar strains

Maria Luisa Ricci[1,10,*,†], Silvia Fillo[2,*,†], Andrea Ciammaruconi[2], Florigio Lista[2], Christophe Ginevra[3,10], Sophie Jarraud[4,10], Antonietta Girolamo[1], Fabrizio Barbanti[1] ⬤, Maria Cristina Rota[1], Diane Lindsay[5,10], Jamie Gorzynski[5], Søren A Uldum[6,10], Sharmin Baig[6], Marina Foti[7], Giancarlo Petralito[2], Stefania Torri[8], Marino Faccini[7], Maira Bonini[7], Gabriella Gentili[7], Sabrina Senatore[7], Anna Lamberti[7], Joao André Carrico[9,10], Maria Scaturro[1,10] ⬤

*Legionella pneumophila* serogroup 1 (Lp1) sequence type (ST) 23 is one of the most commonly detected STs in Italy where it currently causes all investigated outbreaks. ST23 has caused both epidemic and sporadic cases between 1995 and 2018 and was analysed at genomic level and compared with ST23 isolated in other countries to determine possible similarities and differences. A core genome multi-locus sequence typing (cgMLST), based on a previously described set of 1,521 core genes, and single-nucleotide polymorphisms (SNPs) approaches were applied to an ST23 collection including genomes from Italy, France, Denmark and Scotland. DNAs were automatically extracted, libraries prepared using NextEra library kit and MiSeq sequencing performed. Overall, 63 among clinical and environmental Italian Lp1 isolates and a further seven and 11 ST23 from Denmark and Scotland, respectively, were sequenced, and pangenome analysed. Both cgMLST and SNPs analyses showed very few loci and SNP variations in ST23 genomes. All the ST23 causing outbreaks and sporadic cases in Italy and elsewhere, were phylogenetically related independent of year, town or country of isolation. Distances among the ST23s were further shortened when SNPs due to horizontal gene transfers were removed. The Lp1 ST23 isolated in Italy have kept their monophyletic origin, but they are phylogenetically close also to ST23 from other countries. The ST23 are quite widespread in Italy, and a thorough epidemiological investigation is compelled to determine sources of infection when this ST is identified in both LD sporadic cases and outbreaks.

## Introduction

Legionnaires' disease (LD) is a severe pneumonia caused by bacteria belonging to the *Legionella* genus with the most cases due to *Legionella pneumophila* serogroup 1 (Lp1) (Gomez-Valero & Buchrieser, 2019). The illness develops after inhalation of contaminated aerosols mainly produced by different man-made water systems, including showers, cooling towers, and decorative fountains (Chauhan & Shames, 2021). In Europe between 2018 and 2019, the LD incidence was 2.2 per 100,000 inhabitants, showing an increasing trend compared with previous years (European Centre for Disease Prevention and Control, 2020). According to the ECDC surveillance atlas of 2019, Italy reported the highest incidence equal to 5,29 per 100,000 inhabitants and the highest number of LD cases in the EU (n = 3,143). In 2020, the number of cases decreased by 35% compared with 2019, but Italy still had the highest number of reported cases in the EU. The decline might be correlated with the COVID-19 pandemic which led to travel restrictions (fewer travel-associated cases), worldwide government instigated lockdowns and to wearing masks in public areas. (https://atlas.ecdc.europa.eu/public/index.aspx).

Most LD outbreaks are associated with cooling towers because of their ability to spread bacteria over long distances and contaminate other sources (Maisa et al, 2015; Fitzhenry et al, 2017). Genomic matching between clinical and environmental strains is essential for identifying the source of infection. Currently, sequence-based typing (SBT), based on the sequencing of specific regions of seven genes (*flaA*, *pilE*, *asd*, *mip*, *mompS*, *proA* and *neuA*), is the gold standard for genotyping Lp isolates (Gaia et al, 2005).

---

[1]Department of Infectious Diseases Istituto Superiore di Sanità, Rome, Italy   [2]Scientific Department, Army Medical Center, Rome, Italy   [3]CIRI, Centre International de Recherche en Infectiologie, Legionella Pathogenesis Team, University of Lyon, Inserm, U1111, Université Claude Bernard Lyon 1, CNRS, UMR5308, ENS de Lyon, Lyon, France; National Reference Centre of Legionella, Institute of Infectious Agents, Hospices Civils de Lyon, Lyon, France   [4]Université Lyon 1, CNR Legionella, Lyon, France   [5]Scottish Microbiology Reference Laboratories, Glasgow, Scotland   [6]Department of Bacteria, Parasites, and Fungi, Statens Serum Institut, Copenhagen S, Denmark   [7]Agency for Health Protection of Metropolitan Area of Milan (ATS), Milan, Italy   [8]Department of Laboratory of Medicine, Hospital Niguarda, Ca' Granda, Milan, Italy   [9]Faculdade de Medicina, Universidade de Lisboa, Lisbon, Portugal   [10]ESCMID Study Group for Legionella Infections (ESGLI), Basel, Switzerland

Correspondence: maria.scaturro@iss.it
*Maria Luisa Ricci and Silvia Fill contributed equally to this work.
†Maria Luisa Ricci and Silvia Fill are joint first authors.

The ESCMID Study Group for Legionella Infection (ESGLI) has produced a standardized protocol for SBT and many clinical and environmental isolates have been sequence typed and there are currently 14,054 entries and 3,026 STs (https://bioinformatics.phe.org.uk/legionella/legionella_sbt/php/sbt_homepage.php; the data are accessible on request directly to the owner via legionella-sbt@phe.gov.uk). However, SBT alone has not been discriminatory enough in complex outbreak investigations linked to multiple clades or clones of the same ST that were only identified by whole-genome sequencing (WGS) (McAdam et al, 2014; David et al, 2016b).

WGS is now used to investigate LD outbreaks (Schjørring et al, 2017; Besser et al, 2018; Mercante et al, 2018). Mitchell and Simner (2019) showed that WGS gave greater resolution than SBT. This was based on mapping sequence reads to reference genomes followed by the analysis of either single-nucleotide polymorphism (SNP) or a gene-by-gene approach that reproduces on a larger scale the multi-locus sequence typing (MLST). Whole-genome MLST (wgMLST) has been applied to study the genomic diversity of Lp clinical strains, unrelated to specific epidemic events, and interestingly previously unknown clusters were revealed (Raphael et al, 2019). WGS also showed that a single Lp1 clone was the cause of three outbreaks when SNP, core genome MLST and pangenome approach was used (Timms et al, 2018). Recently, a core genome MLST (cgMLST) scheme, based on 1,521 core genes, has been proposed to investigate LD outbreaks (Moran-Gilad et al, 2015) and has been demonstrated to be very useful during LD outbreak investigations. Petzold et al (2017), showed that Lp isolates, sharing a unique ST, that fell into two separate clusters, were only different because of recombination events in specific regions of the genome.

In Bresso, Italy, two outbreaks associated with ST23 were identified in 2014 (6 cases) and in 2018 (52 cases). The only suspected, yet unlikely, source of infection was identified as a public fountain, where Lp1 Philadelphia and Lp1 France/Allentown ST23 matching the clinical isolates was found (Faccini et al, 2020).

The SBT database was queried and ST23 was found to be widespread but in a restricted number of countries including Italy, where it caused all up to now described outbreaks and also several sporadic cases (Rota et al, 2005, 2011; Scaturro et al, 2015). It also represents the most frequently community- and travel-associated ST (Fontana et al, 2014). Therefore, all ST23 isolated from the two outbreaks in Bresso and from the rest of Italy between 1995 and 2018 were analysed and compared with several ST23 isolated from other countries, either present in GenBank or sequenced and de novo assembled as part of this study.

## Results

### Italian isolates, ST, and MAb typing

The monoclonal subgroup of all the Lp 1 isolates analysed by WGS from Italy were MAb 3/1 positive, and they are listed in Table 1. Overall, 58 strains of this collection were ST23 and a mixture of Philadelphia (n = 35), France/Allentown (n = 18) and Knoxville (n = 5). Five isolated in Bresso were a single locus variant of ST23, ST2695, with a single difference seen in the neuA allele (i.e., 37 instead of 6),

and they were Benidorm. They were either community (CA) or travel (TA) acquired but four were of unknown origin. The majority of CA (n = 47) were linked to outbreaks.

### cgMLST analysis

All the 63 Italian ST23/ST2695 were de novo assembled and phylogenetic relationships inferred. First, based on the previously published cgMLST scheme (Moran-Gilad et al, 2015), cgMLST profiles were determined on the basis of the set of 1,362 cgMLST targets shared by the 44 genomes which had caused outbreaks in Italy, including 29 from Bresso, five from Rome, seven from Cesano Maderno, and three from Lazise, and a minimum spanning tree was constructed (Fig 1).

Fig 1 shows three main clades: clade A including 23 of 44 epidemic ST23 showing 1–3 core gene (cg) loci of difference; clade B including five environmental isolates from the Bresso outbreak (ST2695) showing 26 cg loci of difference; clade C including six ST23 isolates from Cesano Maderno outbreak showing 23 cg differences. The 26 cg differences of ST2695 were mainly located in four specific adjoining genomic regions (Table S1). As determined by Geneious analysis, these loci showed an identity of 91–100% and a number of missense nucleotidic substitutions from 0 to 49. Only the lpg0693 locus, encoding a hypothetical protein annotated as LigA interaptin, had a 3-bp insertion and 48 amino acidic substitutions (Table S2). The single-nucleotide variant of neuA allele 37 of ST 2695 differed by only eight nucleotides compared with neuA allele 6 found in ST23.

The differences observed in clade C (Table S3) were in loci adjacent to each other and by Genious analysis, they showed an identity of 90–99%, with missense nucleotidic substitutions ranged from 0 to 24 and the highest number of substitutions was found in the lpg2530 locus, encoding a 3-deoxy-D-arabino-heptulosonate 7-phosphate (DAHP) synthase (Table S4).

Second, a minimum spanning tree was constructed based on the subset of 1,346 core genes shared by all ST23/ST2695 independently on the year and town of isolation and few loci differences were found (Fig 2). Indeed, considering the node representing the genomes 2417, 3933, 383C, 472C (from sporadic cases) and the genome with ID 228C (part of the Bresso outbreak) as the central node, in which the genomes were geographically and temporally very distant, all the others showed between 1 and 60 cg differences. The genome 299C from Verona showed the highest number of cg differences (n = 177). Two further genomes from sporadic cases, one isolated in Rome (1C), the other from Bolzano (325C) diverged at 45 and 60 loci, respectively. However, most of the variations were in contiguous genomic loci, as shown in Tables S5–S7.

To establish if the phylogenetic relationships were found elsewhere, ST23 isolated in other countries were added to the analysis. Danish isolates were sequenced during this study and isolates from Scottish patients that had already been sequenced as part of an ongoing study and others previously reported were included (David et al, 2016a).

The cgMLST based on the subset of 1,216 cgMLST shared by all ST23, as listed in Table S8, were determined (Fig 3). This analysis demonstrated that, independently on the country of isolation (Fig 3A), ST23 differed in a limited number of cg loci, ranging from 1 to 158. The highest number of cg differences was found in 299C from

**Table 1.** *Legionella pneumophila* serogroup 1 strains analysed by whole genome sequencing.

| Strain ID | Source | Origin | Year isolation | Investigation context | ST | Monoclonal subgroup |
|---|---|---|---|---|---|---|
| 228C | Clinical | | 2014 | CA-O | 23 | Philadelphia |
| 2251B | Environmental | | 2014 | CA-O | 23 | France Allentown |
| 2251C | Environmental | Bresso outbreak | 2014 | CA-O | 23 | France Allentown |
| 2251D | Environmental | | 2014 | CA-O | 23 | France Allentown |
| 427C | Clinical | | 2018 | CA-O | 23 | Philadelphia |
| 428C | Clinical | | 2018 | CA-O | 23 | Philadelphia |
| 435C | Clinical | | 2018 | CA-O | 23 | Philadelphia |
| 436C | Clinical | | 2018 | CA-O | 23 | Philadelphia |
| 2227A | Environmental | | 2018 | CA-O | 23 | France Allentown |
| 2252A | Environmental | | 2018 | CA-O | 2695 | Benidorm |
| 2252C | Environmental | | 2018 | CA-O | 2695 | Benidorm |
| 2253A | Environmental | | 2018 | CA-O | 2695 | Benidorm |
| 2253B | Environmental | | 2018 | CA-O | 2695 | Benidorm |
| 2253C | Environmental | | 2018 | CA-O | 2695 | Benidorm |
| 2255A | Environmental | | 2018 | CA-O | 23 | France Allentown |
| 2256A | Environmental | | 2018 | CA-O | 23 | France Allentown |
| 2257A | Environmental | Bresso outbreak | 2018 | CA-O | 23 | France Allentown |
| 2258A | Environmental | | 2018 | CA-O | 23 | France Allentown |
| 2259A | Environmental | | 2018 | CA-O | 23 | France Allentown |
| 2260A | Environmental | | 2018 | CA-O | 23 | France Allentown |
| 2261A | Environmental | | 2018 | CA-O | 23 | France Allentown |
| 2452B1 | Environmental | | 2018 | CA-O | 23 | Philadelphia |
| 2452B2 | Environmental | | 2018 | CA-O | 23 | Philadelphia |
| 2452C1 | Environmental | | 2018 | CA-O | 23 | Philadelphia |
| 2452C2 | Environmental | | 2018 | CA-O | 23 | Philadelphia |
| 2452D1 | Environmental | | 2018 | CA-O | 23 | Philadelphia |
| 2452D2 | Environmental | | 2018 | CA-O | 23 | Philadelphia |
| 2452A1 | Environmental | | 2018 | CA-O | 23 | Philadelphia |
| 2452A2 | Environmental | | 2018 | CA-O | 23 | Philadelphia |
| 3699 | Clinical | | 2003 | CA-O | 23 | Philadelphia |
| 3712 | Environmental | | 2003 | CA-O | 23 | Philadelphia |
| 3718 | Environmental | Rome outbreak | 2003 | CA-O | 23 | Philadelphia |
| 3713 | Environmental | | 2003 | CA-O | 23 | Philadelphia |
| 3777 | Environmental | | 2003 | CA-O | 23 | Philadelphia |
| 4454 | Clinical | | 2007 | CA-O | 23 | Knoxville |
| 18C | Clinical | | 2008 | CA-O | 23 | Philadelphia |
| 4407 | Environmental | | 2007 | CA-O | 23 | Knoxville |
| 22A | Environmental | Cesano Maderno outbreak | 2007 | CA-O | 23 | Knoxville |
| 11A | Environmental | | 2007 | CA-O | 23 | Philadelphia |
| 151A | Environmental | | 2007 | CA-O | 23 | Knoxville |
| 160A | Environmental | | 2008 | CA-O | 23 | Philadelphia |

**Table 1.  Continued**

| Strain ID | Source | Origin | Year isolation | Investigation context | ST | Monoclonal subgroup |
|---|---|---|---|---|---|---|
| 143C | Clinical | | 2011 | TA-O | 23 | France Allentown |
| 594A | Environmental | Lazise outbreak | 2011 | TA-O | 23 | France Allentown |
| 595A | Environmental | | 2011 | TA-O | 23 | France Allentown |
| 2417 | Clinical | Monza | 1995 | Unknown-SP | 23 | Philadelphia |
| 2418 | Clinical | Monza | 1995 | Unknown-SP | 23 | Philadelphia |
| 3933 | Clinical | Trento | 2004 | CA-SP | 23 | Philadelphia |
| 1C | Clinical | Roma | 2007 | CA-SP | 23 | Knoxville |
| 299C | Clinical | Verona | 2011 | Unknown-SP | 23 | Philadelphia |
| 300C | Clinical | Verona | 2011 | Unknown-SP | 23 | Philadelphia |
| 717A | Environmental | Ravenna | 2012 | TA-O | 23 | France Allentown |
| 181C | Clinical | Ravenna | 2012 | TA-O | 23 | France Allentown |
| 1214A | Environmental | Piacenza | 2015 | CA-O | 23 | Philadelphia |
| 323C | Clinical | Cesena | 2016 | CA-SP | 23 | Philadelphia |
| 325C | Clinical | Bolzano | 2017 | TA-SP | 23 | Philadelphia |
| 383C | Clinical | Ancona | 2017 | Unknown-SP | 23 | Philadelphia |
| 384C | Clinical | Como | 2017 | CA-O | 23 | Philadelphia |
| 415C | Clinical | Milano | 2017 | CA-SP | 23 | Philadelphia |
| 419C | Clinical | Bolzano | 2017 | CA-SP | 23 | Philadelphia |
| 1762A | Environmental | Como | 2017 | CA-O | 23 | Philadelphia |
| 472C | Clinical | Milano | 2018 | CA-O | 23 | Philadelphia |
| 483C | Clinical | Mantova | 2018 | CA-O | 23 | France Allentown |
| 2301A | Environmental | Brescia | 2018 | CA-O | 23 | France Allentown |

CA-O, community-acquired outbreak; TA-O, travel-associated outbreak; CA-SP, community-acquired sporadic; TA-SP, travel-associated sporadic. A bold line divides epidemic from sporadic Lp1 strains; a thinner line highlights the different Italian outbreaks.

Verona. The ST23 isolated from Danish patients (orange nodes in Fig 3A) differed for 1–28 loci from ST23 isolated in Italy, France, South Europe, and Switzerland. In particular, the genomes having DK02 and DK03, from a Danish patient, showed only 12 cg loci differences encoding Icm proteins of the Dot/Icm type IV secretion system, but as determined by Geneious analysis the variations were all conserved.

ST23 isolated in Scotland from patients who had travelled to Southern Europe differed by between 1 and 33 cg loci from ST23 isolated in Italy and France. Generally, the loci differences found in ST23 genomes mainly corresponded to genes encoding proteins related to the carbohydrate, amino acids, and lipid metabolism, and several for transport and binding (Table S9). As expected, independently on the country of isolation, all the ST23 were quite distant from other common STs, such as ST1, ST37, and ST62 (Fig 3B).

### SNPs analysis

SNP analysis was used to provide phylogenetic information about ST23. SNP analysis performed on the 63 ST23 (Table 1) and other ST23 collected during this study (Table S10) provided an 8,323 and 10,421 core SNP, respectively.

Fig 4 represents a minimum spanning tree of the 8,323 core SNP of the Italian data set with maximum pairwise SNP differences ranged from 1 to 2,442, and the highest number of SNP was found in

299C from Verona, Italy, (Fig 4 and Table S7). If comparing all ST23 strains with older isolates 2,417, Monza (year = 1995) and EUL 41 (Italy, year = 1999), both located in the central node, the majority of strains differed by very few SNP. Forty-three isolates had SNPs ranging between 1 and 40, including those from Bresso, Lazise, and Rome outbreaks, from sporadic cases and EUL8 and EUL11, both isolated in Switzerland (Table S11 and Fig 4). The EUL4 from Switzerland was the most distant with 1,778 SNPs. 21 genomes showed a higher number of SNP ranging from 171 to 802 (Table S12).

The origin of these variations determined by Gubbins analysis showed that 99.7% of the SNP were a result of recombination. After removing the recombined regions, corresponding to a mean of 3.13% of the 3.3 Mb genomes, to leave only the SNP vertically transmitted, the maximum number of pairwise SNP differences was 48 (found in 299C) with a mean of 6.7 for all ST23.

A 10,421 core SNP of all 120 genomes, including this entire collection (ST23/ST2695) compared with ST1, ST37, and ST62 showed 2,602, 7,265, and 1,745 SNP differences, respectively, whereas among the ST23, the highest number of SNPs (392) was found in the genome 299C (Fig 5).

Overall SNP analysis provided greater resolution than cgMLST. The minimum spanning tree generated from core genome SNP showed the central node, consisting of five strains, further separated into five different nodes. SNP analysis identified six polymorphisms (pairwise 2,417–3,933) among strains undistinguishable by cgMLST.

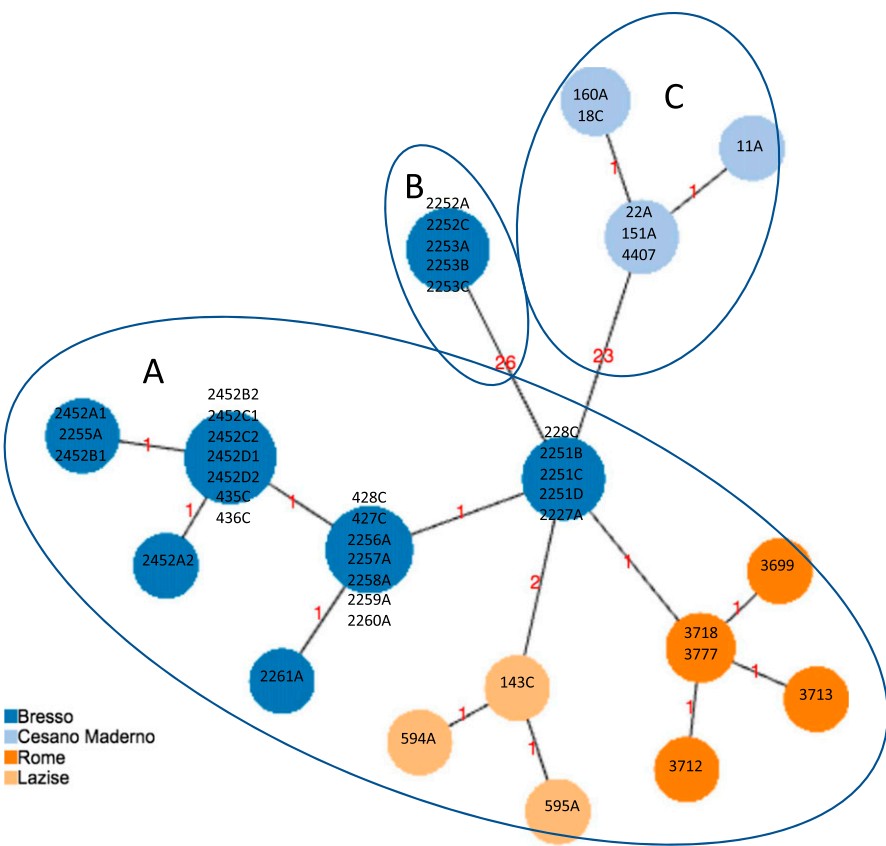

**Figure 1.   cgMLST of the Italian ST23 epidemic genomes.**
Minimum spanning tree based on the set of 1,362 cgMLST targets of the *Legionella pneumophila* ST23/ST2695 genomes, causing epidemic events in Italy, is shown. The isolates from Bresso 2014 and 2018, Rome, and Lazise outbreaks differed for 1, 2 or 3 loci of difference (clade A); in clade B, the five genomes from Bresso 2018 typed ST2695 with 26 loci of difference; in clade C, genomes from Cesano Maderno outbreak showed 23 loci of difference from the A clade. The clades are evidenced by circles.

To verify the origin of SNP in this collection, ST23 were aligned against a reference genome using snippy and the maximum pairwise SNP difference calculated for the entire ST23 collection was 4,799 and was found in the HL05063005 genome from France (Table S8). As determined by Gubbins, a range of 37.5–99.9% of SNP differences were due to recombination. The genomes that showed least recombinations were those from (DK10) and Southern Europe (4697). After removal of the SNP due to recombination, which represented a mean of 4.62%, the maximum pairwise differences was 46 and again it was found in 299C genome (Fig S1).

### Pangenome

As determined by Roary pipeline, the pangenome consisted of 2,662 core genes and 1,511 accessory genes out of a total of 4,675 genes (including 280 soft-core genes and 222 shell not counted in the core and accessory genome). As shown in Fig 6A–D, accessory genome consisted of genes shared by unrelated ST23 (red rectangle) and genes present only in closely related ST23 (blue rectangle).

## Discussion

Lp ST23 are a well-known cause of both sporadic and epidemic LD cases, representing 4.4% of the ST found in Italy and 6.6% of the ST

reported to the SBT database, although reporting countries are only France and Italy, with France having the most abundant number of ST23 cases (Sánchez-Busó et al, 2016; Zanella et al, 2018).

In this study, the ST23 collection investigated was found to be quite preserved which is in contrast to the genomic plasticity and variation observed with other *L. pneumophila* and among diverse *Legionella* species (Cazalet et al, 2004, 2008), mostly due to the presence of mobile genetic elements such as transposons and insertion sequences, recombination regions, and conjugative elements.

The year and geographic area of isolation for all Italian ST23 showed highly similar cgMLST and a very restricted number of SNP differences, which suggests that ST23 has undergone a very low selective pressure, according to a recent appearance and an adjustment to a specific environmental niche (David et al, 2016a). However, when all Italian ST23s were compared with those of other countries, having both a high and low incidence of ST23 cases, such as France and Denmark, respectively, cgMLST and SNPs analyses again evidenced a close phylogenetic relation, and ST23s were not country specific.

The phylogenetic relation was better highlighted when the entire ST23 collection subject of this study was compared with other STs, such as ST1, 37, and 62. Even with the small sample size, the non-ST23 showed the highest variability for both number of cg loci and SNP differences.

It is noteworthy that, although the ST23 is found, for example, in Denmark, the cases are mainly travel-associated and imported

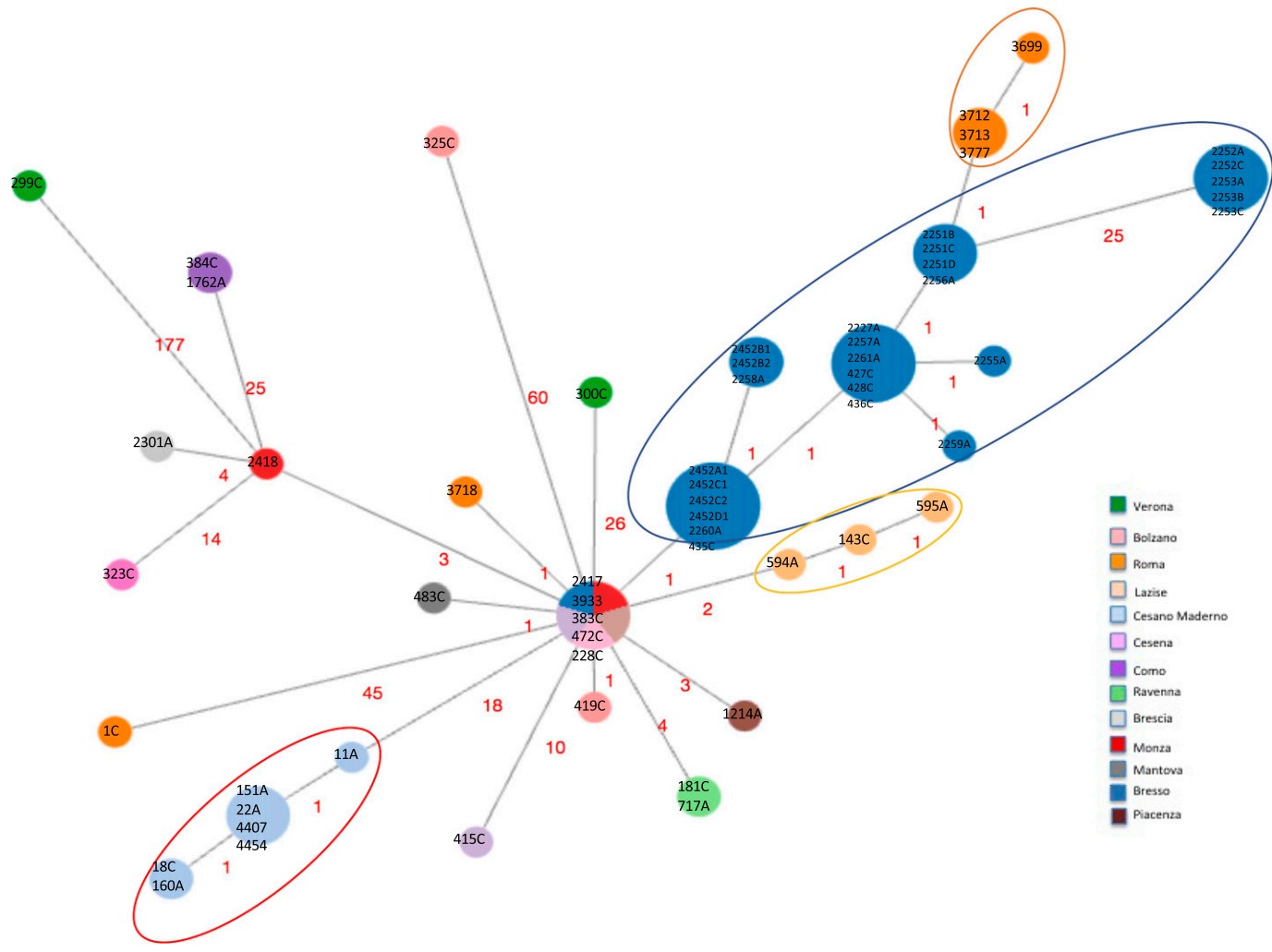

**Figure 2. cgMLST of the Italian ST23 genomes that caused both epidemic and sporadic LD cases.**
Minimum spanning tree based on the set of 1,346 cgMLST targets shared by the 63 Italian genomes is shown. Nodes are proportionated to the number of genomes included and colors correspond to the towns of isolation. Epidemic genomes are evidenced by circles. The number of loci of difference is indicated in the links. The central node included four genomes causing sporadic cases, one from Monza (ID strain: 2417, year 1995), one from Trento (ID strain: 3933, year 2004), one from Ancona (ID strain: 383C, year 2017), one from Milan (ID strain: 472C, year 2018), and one epidemic isolated during the Bresso 2014 outbreak (ID strain: 228C). Sporadic cases with the corresponding environmental related are found in the same node: 384C with 1762A from Como and 181C with 717A from Ravenna.

primarily from Central/South Europe, but not from France. This observation could be interpreted as a restricted geographic distribution of ST23 (probably still spreading), and the geographical distribution of cases would not represent the real areal circulation of this linage. On the contrary, it could be speculated that the genomes DK04 and DK05, both associated with a resort in south Sweden and indistinguishable from Italian strains, could be believed as a recent introduction (year of isolation 2017) in the north Europe, incoming from Italy via different spreading mechanisms, such as wind transport.

At intra-ST23 level, most loci differences were located adjacent to each other and upon removing of horizontally transmitted SNP due to recombination events, the distances among the ST23 were significantly reduced, suggesting that the SNP differences observed were mainly due to recombination events irrespective of the year or country of isolation. This behaviour has already been observed in

ST23 and in ST1, ST37, and ST62 but not in ST47, where recombinations events were not found (David et al, 2016a). It has been described that regions encompassing several genes, with a high number of polymorphisms, have exchanged DNA with different strains and species, and this phenomen is more frequently found in strains isolated in the environment (Gomez-Valero et al, 2011). Consisting with the "dead-end" nature of *Legionella* infection, only the environment can exert a selective pressure. Therefore, it appears surprising that genomes isolated in different countries could be so phylogenetically close. The pangenome data highlighted the presence of accessory genes shared by unrelated ST23, but also specifically present in ST23 closely related in terms of country of origin. This pool of accessory genes could have been determined by plasmids and other mobile genetic elements that have not contributed to divergence of the ST23 genomes. Understanding the origin of these genes will be the subject of further investigations,

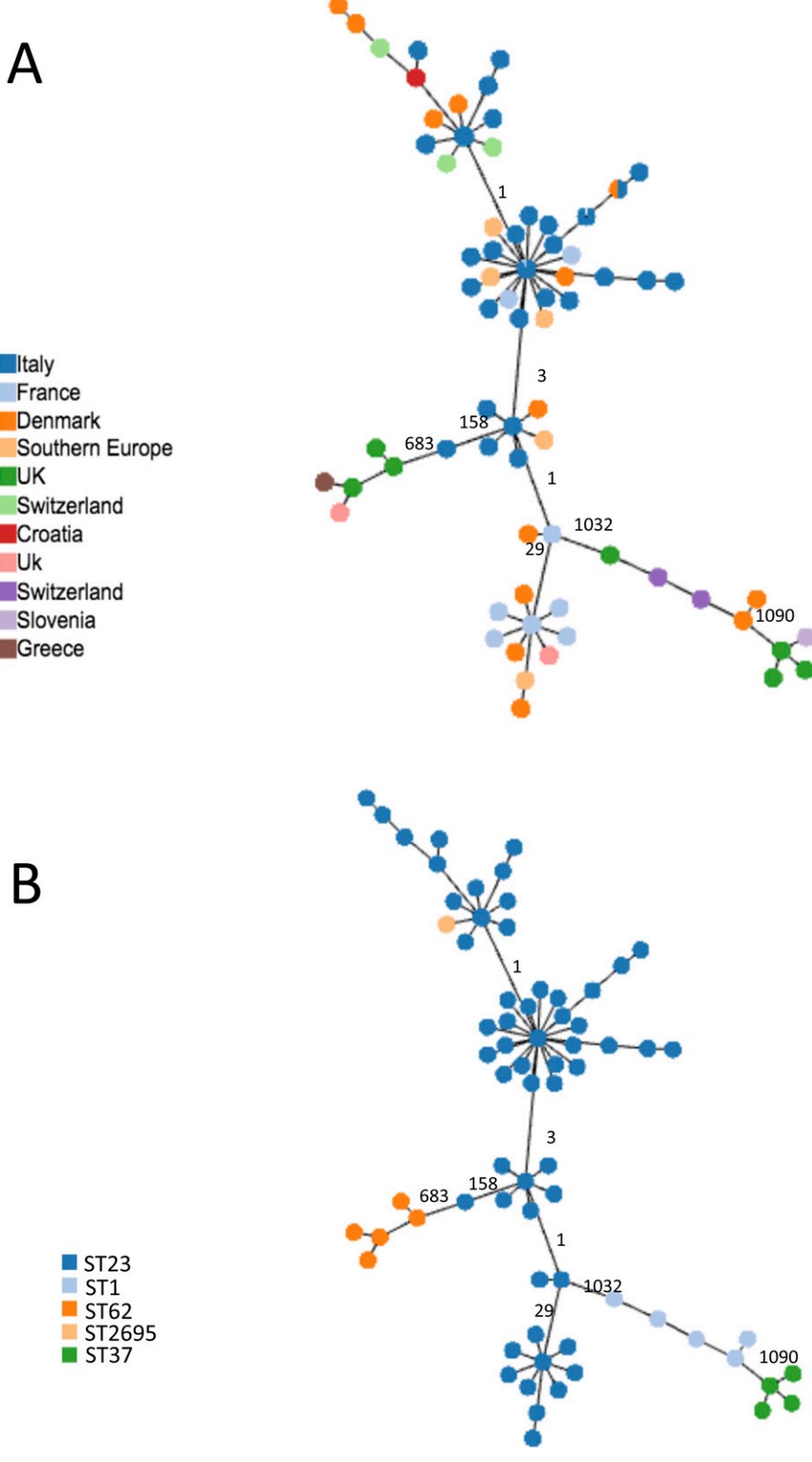

**Figure 3. cgMLST of ST23 isolated in Italy and in other countries.**
Minimum spanning tree based on the set of 1,216 cgMLST targets shared by all the ST23 object of this study and by 14 not-ST23 is shown. **(A, B)** Colors represent the countries where genomes were isolated; (B) colors represent the STs. In (A), UK and Switzerland are represented by two colors because belonging to different STs. In (A, B), the numbers of loci of difference in the principal clades are also indicated on the link.

A

**Legend (A):**
- Italy
- France
- Denmark
- Southern Europe
- UK
- Switzerland
- Croatia
- Uk
- Switzerland
- Slovenia
- Greece

B

**Legend (B):**
- ST23
- ST1
- ST62
- ST2695
- ST37

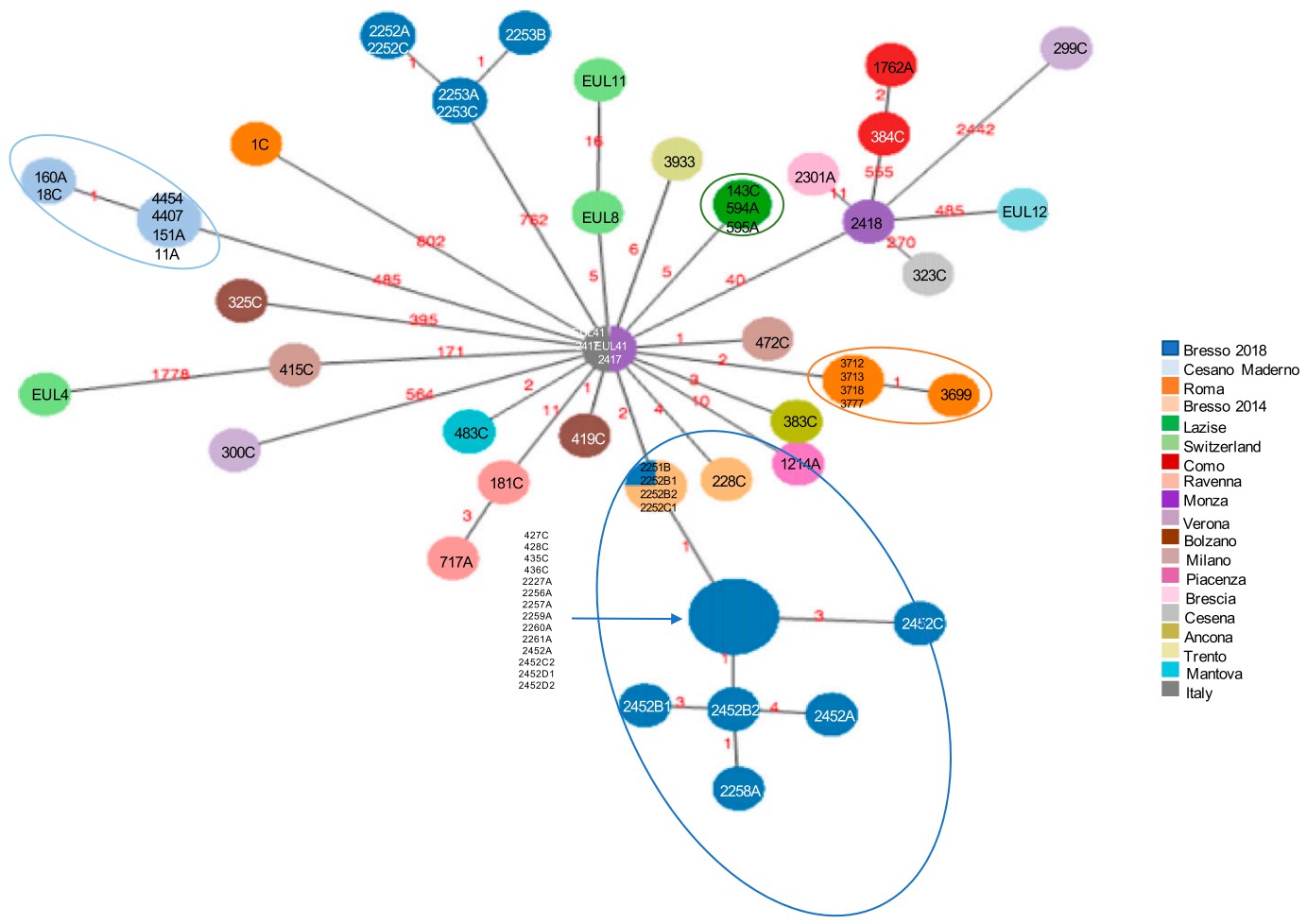

**Figure 4. Single-nucleotide polymorphism (SNP) analysis of the Italian ST23.**
Minimum spanning tree of the 8,323 core SNPs obtained by kSNP analysis is shown. Nodes of different sizes represent a different number of genomes they include and circles highlight the epidemic genomes. Genomes ID are indicated and colors correspond to the towns where they were isolated. This analysis also included EUL reference strains: EUL12 and EUL 41 isolated in Italy, and EUL4, EUL8, and EUL11 isolated in Switzerland. SNPs are also indicated on the links.

possibly including a larger number of genomes from other countries. However, it could be speculated that the recombination events could have introduced some advantages for *Legionella* pathogenesis, as determined by the major spread of ST23 among community- and travel-acquired cases.

This study demonstrated that the ST23 responsible for the two outbreaks occurred in Bresso in 2014 and 2018 were genetically very similar. In 2018, during the outbreak, epidemiological and microbiological data identified a decorative fountain located in a public garden as a probable source of infection (Faccini et al, 2020). cgMLST clearly showed a strong phylogenetic correlation between clinical and environmental isolates and concluded that the fountain did contain the epidemic ST23 clone, but was unlikely to be the sole source of infection to cause such a large outbreak and that the epidemic clone may be endemic within the Bresso water systems but was either not tested or had been treated before testing.

In Bresso, ST2695, a single locus variant of ST23 was also isolated and this also showed nucleotide variations in neighbouring loci,

attributable to recombination events and when SNP due to horizontal transfer was removed, the divergence was further reduced. Therefore, it is possible that the newer ST2695 could have originated from ST23 and belongs to the same linage. Additional information about the ST23 phylogeny could be provided by other strains with single allele differences from ST23, which were unfortunately not isolated.

In two of the most diverse ST23 (299C and 325C), a region of about 20 Kb was identified encoding the putative virulence gene *mvi*N and *enh*A/C involved in uptake into host cells (Liu et al, 2008). These genes share the same amino acid substitutions, like a common event could have been interested in both genomes. In addition, 299C showed a 7-Kb region of variation where flagellar biosynthesis genes are located. In general, the other differences found mainly concern genes encoding hypothetical proteins or metabolic enzymes. For example, in ST2695 differences in heat-shock proteins, such as groES/EL, and the N-acetylneuraminic acid synthase neuB were highlighted, whereas in ST23 from Cesano Maderno the legA-14 belonging to the ankyrin repeat-containing family proteins Dot/

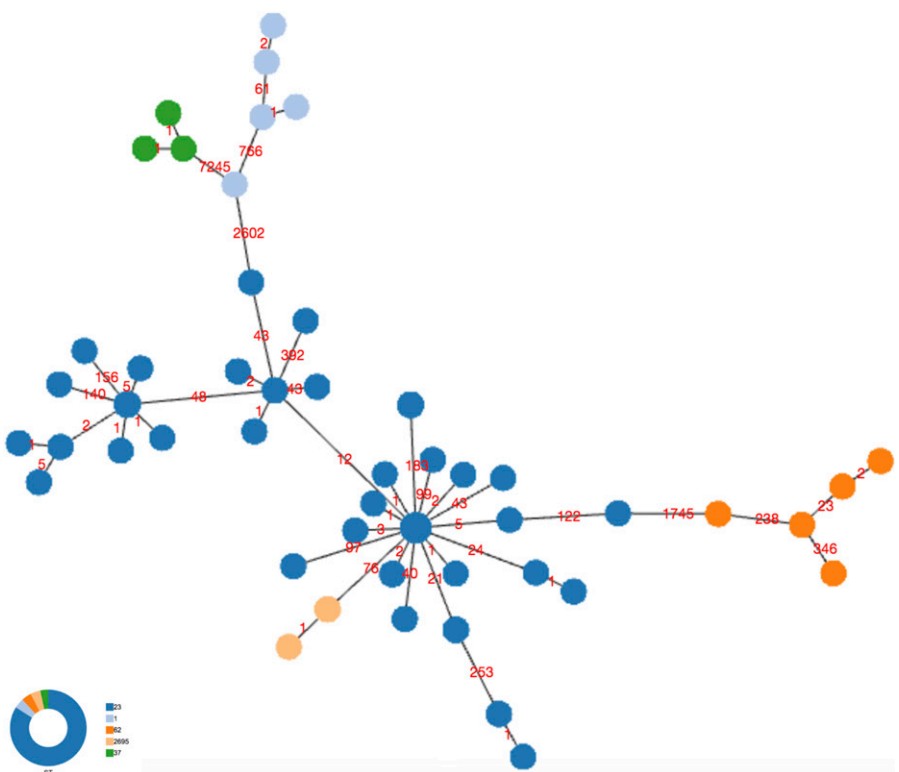

**Figure 5. Single-nucleotide polymorphism (SNP) analysis of the ST23 and not-ST23.**
Minimum spanning tree of the 10,421 core SNPs obtained by kSNPs analysis of the entire collection of ST23 and not-ST23, object of this study, is shown. Colors represent the STs, and number of SNPs is indicated on the links.

Icm-translocated involved in intracellular proliferation (Al-Khodor et al, 2008) were also found. Looking at the entire ST23 collection here investigated, it was interesting to observe a genomic region of variation of about 20 Kb, involving the genomes 2252A, DK02, DK03, and EUL4, containing 11 *dot/icm* genes, well known to be responsible of the *Legionella* pathogenesis.

It is noteworthy that, apart the ST23 isolated in France, most of the ST23s of other countries were from travel-associated cases and therefore not representative of an evolutive pressure determined by a specific geographic niche. ST23 isolated in both Denmark and Scotland, both countries with a low incidence of ST23 LD cases, were all from abroad travel-associated cases. On the contrary, French ST23, mostly from national community-acquired cases, were a good example of a genomic similarity from a large, single geographical area. In the Italian ST23 LD cases, southern Italy was not represented and this is could be probably due to a lower notification rate compared with the North of the country. Therefore, the Italian ST23 are from an even smaller geographic region of a country with high ST23 incidence and high genomic similarity.

Looking at the ST23 isolated in Italy and those entered the SBT database, it is noteworthy that they are prevalently associated to travel- and community-acquired LD cases. The evidence that the ST23 strains are able to cause disease in a population not at high risk of infection deserves more in-depth studies, with particular regard to its virulence features. It would also be interesting to further investigate the presence of environmental ST23 strains, to identify specific water ecological niches and adopt appropriate prevention and control measures to limit future outbreaks.

## Conclusion

This is the first study focusing on Lp ST23 showing that it is highly conserved although distant temporally and geographically. It also highlights the importance of combining both epidemiological and genomic data, adopting typing methods with highly discriminatory power, to identify new clusters and the possible source of infection for future LD incidents caused by this ST.

# Materials and Methods

### Bacterial strains and genomes

In total, 63 *L. pneumophila* ST23/ST2695 from the collection of the National reference laboratory for *Legionella*, Rome, Italy were analysed: 29 of them were from the two Bresso outbreaks in 2014 and 2018, 15 were from outbreaks occurred in Rome, Cesano Maderno, and Lazise, whereas 19 were from sporadic cases. These last included 15 clinical and four environmental strains, isolated between 1995 and 2018 in different Italian towns. They were all ST23 except five that were ST2695, differing from ST23 for just the neuA allele, and they were all isolated following either community- or travel-associated outbreaks and sporadic LD cases, as described in Table 1. This Italian ST23 collection analysed here was chosen to be as representative as possible of Italy although few isolates were from the South. All genomes were submitted to GenBank (Bioproject PRJNA606275; accession numbers listed in Table S8).

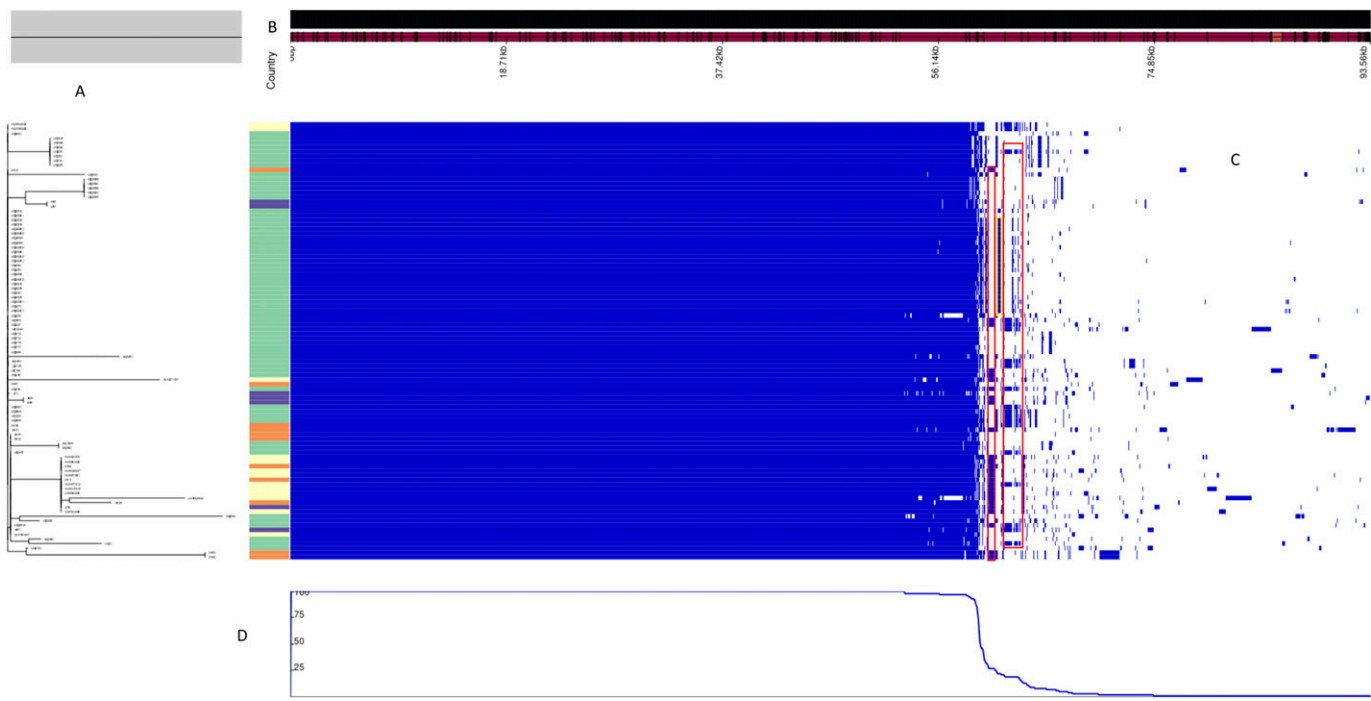

**Figure 6. Pangenome of the ST23 data set (n = 96).**
**(A)** Maximum likelihood tree showing the grouping of the ST23s colored column indicating the countries of origin: light yellow is France, green is Italy, orange is Denmark, violet is South Europe. **(B)** Pangenome sorted from core genes on the left to accessory genes on the right. **(C)** Heat map showing presence (blue) and absence (white) of genomic regions. Red squares highlight accessory genes shared by ST23 from different countries; yellow square highlights genes present only in ST23 from Italy. **(D)** Gene frequency.

To better understand the phylogenies of ST23 isolated in Italy, further 41 ST23 isolates already published, sequenced elsewhere or as part of this study were included. In addition, 14 genomes, ST1 (n = 5), ST37 (n = 4), and ST62 (n = 5) were also included (Table S10).

### Culture and DNA extraction of Lp1 isolates

Lp 1 strains stored at −80°C were thawed and cultured on Buffered Charcoal Yeast Extract (BCYE, Oxoid) agar at 37°C for 48 h. A suspension on 1% formalin was used for MAb typing by indirect immune-fluorescence according to Dresden method (Helbig et al, 1997, 2002). Isolated colonies were suspended in PCR-grade water for automated DNA extraction using QIAcube with QIAmp mini kit (QIAGEN). The DNA extracts were used for next generation sequencing and SBT (Gaia et al, 2005).

### WGS and assembly

Sequencing libraries were prepared using the NextEra XT library prep kit (Illumina) and then run on two different Illumina platforms: a 150-bp paired-end sequencing run was performed on the NextSeq-500 (Illumina) using a Mid output Kit v2, whereas a 250-bp paired-end sequencing run was performed on the Miseq (Illumina). The fastq files were trimmed and assembled using the fully automated pipeline INNUca by Galaxy ARIES platform (https://aries.iss.it/; Mitchell & Simner, 2019).

Eleven ST23 from Danish patients were sequenced at the Statens Serum Institut, as described above, and the reads were de novo assembled using SPAdes v3.12.1 with default parameters (Bankevich et al, 2012) (Bioproject PRJEB48072). A further seven ST23, originated from Scottish patients who had travelled to the Southern Europe, were sequenced at the University of Birmingham by MicrobesNG, using Illumina 2 × 250 bp paired-end short read sequencing and deposited in the SRA database (Submission ERA1765603: Bioproject PRJEB31628).

### cgMLST analysis

chewBBACA software (24–25) was used to convert the cgMLST scheme defined by Moran-Gilad et al (2015), which included 1,521 core genes. The allelic profiles were then corrected to include only the subset of genes that were present in all the strains. The allelic profile output was then used to create minimum spanning trees with PHYLOViZ Online software (Ribeiro-Gonçalves et al, 2016; Silva et al, 2018; Knijn et al, 2020 Preprint).

Geneious 9.1.8 (Biomatters Ltd.) was used for nucleotide sequence alignment of loci to identify differences against corresponding nucleotide reference sequences.

### SNP analysis

The kSNP program, based on k-mer analysis, was used to identify core genome SNP on WGS data for each isolate. After the kSNP

analysis, the core SNP matrix output file, containing only SNP loci common to all isolates, was used to construct minimum spanning tree by PHYLOViZ online software.

To investigate if SNP occurrence was due to recombination events, sequence reads were mapped to the LT632615.1 ST23 reference genome using Snippy v4.5.0, and horizontally transmitted SNP were removed by Gubbins analysis (https://aries.iss.it/; Mitchell & Simner, 2019). Tree was inferred using the Neighbor-Joining method (Saitou & Nei, 1987) and the evolutionary distances were computed using the Maximum Composite Likelihood method (Tamura et al, 2004) that were in the units of the number of base substitutions per site. All ambiguous positions were removed for each sequence pair (pairwise deletion option). Evolutionary analyses were conducted in MEGA X (Kumar et al, 2018).

### Pangenome

Genomes were annotated using Prokka (Seemann 2014) and pangenome analyses were performed using Roary (Page et al, 2015). The phylogenetic tree used as input for Roary was based on core SNP and was calculated using Parsnp (Treangen et al, 2014).

# Data Availability

All raw and processed sequencing data generated in this study have been submitted to the National Center for Biotechnology Information (NCBI; https://www.ncbi.nlm.nih.gov/) Bioproject database PRJNA606275 and accession numbers are also listed in Table S10 in Supplemental material. All the other isolates, sequenced in this study, were submitted to bioprojects PRJEB48072 and PRJEB31628.

# Supplementary Information

# Acknowledgements

The authors are grateful to all health professionals (microbiologists, doctors, nurses, health assistants, laboratory technicians, etc.) who have collaborated by sending the surveillance forms, respiratory samples and isolated *Legionella* strains. The authors thank Antonella Fortunato for technical support. This work was supported by the Ministry of Health (Centro per il controllo delle malattie, 2019-2020; Grant ISS N. R934). We thank Anna Maria Marella for her technical support.

## Author Contributions

ML Ricci: conceptualization, formal analysis, and investigation.
S Fillo: conceptualization, formal analysis, and investigation.
A Ciammaruconi: software, investigation, and methodology.
F Lista: formal analysis.
C Ginevra: software and methodology.
S Jarraud: supervision.
A Girolamo: investigation and methodology.
F Barbanti: investigation and methodology.
MC Rota: supervision and investigation.
D Lindsay: investigation and writing—review and editing.
J Gorzynski: methodology.
SA Uldum: supervision and methodology.
S Baig: methodology.
M Foti: methodology.
G Petralito: methodology.
S Torri: methodology.
M Faccini: supervision.
M Bonini: methodology.
G Gentili: methodology.
S Senatore: supervision.
A Lamberti: methodology.
JA Carrico: software, formal analysis, supervision, and methodology.
M Scaturro: conceptualization, data curation, software, investigation, and writing—original draft, review, and editing.

## Conflict of Interest Statement

The authors declare that they have no conflict of interest.

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
