## [Reviewer comments · Life Science Alliance]

Life Science Alliance

Genome analysis of *Legionella pneumophila* ST23 from various countries reveals highly similar strains

Maria Luisa Ricci, Silvia Fillo, Andrea Ciammaruconi, Florigio Lista, Christophe Ginevra, Sophie Jarraud, Antonietta Girolamo, Fabrizio Barbanti, Maria Cristina Rota, Diane Lindsay, Jamie Gorzynski, Søren Uldum, Sharmin Baig, Marina Foti, Giancarlo Petralito, Stefania Torri, Marino Faccini, Maira Bonini, Gabriella Gentili, Sabrina Senatore, Anna Lamberti, Joao Andre' Carrico and Maria Scaturro

DOI: <https://doi.org/10.26508/lsa.202101117>

Corresponding author(s): Dr. Maria Scaturro (Istituto Superiore di Sanità)

Review Timeline:

Submission Date:	2021-05-11
Editorial Decision:	2021-07-15
Revision Received:	2022-01-14
Editorial Decision:	2022-02-08
Revision Received:	2022-02-14
Accepted:	2022-02-15

Scientific Editor: Novella Guidi

Transaction Report:

July 15, 2021

Re: Life Science Alliance manuscript #LSA-2021-01117-T

Maria Scaturro
Istituto Superiore di Sanit 

Dear Dr. Scaturro,

Thank you for submitting your manuscript entitled "Genome analysis of Legionella pneumophila ST23, responsible for sporadic and epidemic cases in Italy, 1995-2018, reveals highly conservative strains." to Life Science Alliance. The manuscript was assessed by expert reviewers, whose comments are appended to this letter. As you will note from the reviewers' comments below, both reviewer are quite positive and excited about the work that in their view provides new evidence of new genomic sequences of a pathogen of interest in Italy, Europe and also worldwide. They just raised few minor comments regarding the analysis conducted that need to be addressed in the manuscript. We, thus, encourage you to submit a revised version of the manuscript back to LSA that responds to all of the reviewers' points including revising taxon sampling and phylogenetic analyses as suggested by reviewer 1. We also encourage you, in line with reviewer 1, to run some genomic comparisons as to compare gains and losses of the isolates of interest, as well as differences in gene complement which can be connected to virulence or survival. This analysis will clearly strengthen the findings.

Thank you for this interesting contribution to Life Science Alliance. We are looking forward to receiving your revised manuscript.

Sincerely,

B. MANUSCRIPT ORGANIZATION AND FORMATTING:

Reviewer #1 (Comments to the Authors (Required)):

The manuscript submitted by Ricci et al. examines the genomic sequences of several dozens of the bacteria 'Legionella pneumophila' serogroup 1 (Lp1) sequence type (ST) 23, obtained from clinical and environmental isolates, and responsible for epidemic and sporadic cases of Legionary Disease between 1995 and 2018 in Italy. The authors observe that all of the isolates are closely related, with little variation, mainly due to recombination events. The authors then conclude a clonal origin of the Italian ST23 type.

I think this work has merit. It produces 62 new genomic sequences of a pathogen of interest in Italy, Europe and also worldwide. The general approach of genomic epidemiology is proving to be very fruitful as it develops very swiftly. Nevertheless, I have some major concerns about the extent of the validity of the general conclusions of this manuscript. My doubts are mainly driven by issues regarding taxon sampling and phylogenetic analyses. I will explain myself.

First, concerning the taxon sampling issue, more genomic sequences of Lp1 ST23, and probably ST2695 and other STs, should be included in the analyses. Indeed, the variability observed within the Italian genome sequences is as high as the observed between some Italian and the other European sequences examined in this study (Sup. Table 1). Hence, a more thorough taxon sampling should be carried out in order to adequately assess variability, transmissions and origins of the Italian variants. These data can be obtained from studies as David et al., 2017 (<https://doi.org/10.1371/journal.pgen.1006855>) or crossing databases as PATRIC (<https://www.patricbrc.org/>) and LegionellaDB (<https://doi.org/10.1016/j.tim.2021.01.015>).

In regards to the phylogenetic analyses, the phylogenetic tree from figure 4 is not appropriately presented. First, the tree inference method is not described. Second, no EUL000* data was presented. A new tree rooted and inferred correctly should be done including the newly selected genomes and a potential outgroup. In any case, David et al (2017) mentioned above show a clear methodology on how to analyse SNP data. In addition, the color code of geographic regions should be maintained in the wgMLST and also in the SNP analyses. It could be helpful to see the year of isolation, which could be included in the tip labels.

With these new results at hand, it would be important to compare wgMLST and SNP phylogenetic approaches and discuss the eventual differences between them.

Finally I think that, given the amount of data generated, deeper analyses could be carried out. In addition to the anecdotal description of the differences in alleles some genomic comparisons could be done so as to compare gains and losses of the isolates of interest, as well as differences in gene complement which can be connected to virulence or survival. To perform this analyses inhouse could be somewhat demanding of bioinformatic resources; however, reasonable and meaningful results of "blast atlases" and "core" and "accessory" genomes can be obtained with online servers such as GView (<https://server.gview.ca/>).

There are many typos in the manuscript and I think it could benefit from an editing process.

Summarizing, in my opinion the data presented in this work is valuable but needs a more thorough analysis to clarify certain claims of the article, and also to complement the descriptive work of the observed natural variation of this pathogen connected to clinical and environmental conditions in epidemic and sporadic contexts.

Reviewer #2 (Comments to the Authors (Required)):

This work sought to analyze the Legionella pneumophila serogroup 1 (Lp1) sequence type (ST) 23 at genomic level. A core genome multi locus sequence typing (cgMLST), based on a previously described set of 1521 core genes, and single nucleotide polymorphisms (SNPs) approaches were applied to the Lp1 ST23 strain collection. Although very few loci and single nucleotide variations occurring in ST23 genomes, they found that most of the variations were located in adjacent genomic regions due to recombination events. The evidence is sound, and the conclusion is well supported by the data presented in the work

I only have several minor comments for discussion of this work:

1. It is not surprised to see that very few variations were found in ST23 genomes. However, it would be interesting to investigate or, at least, to discuss whether there are any genotype hot spot(s) related with the drug-resistance of LP?
2. They found that "the 99.7% of SNP had been acquired by recombination". Can authors discuss a little bit more why LP acquire most of SNPs via recombination?
3. They found that "most of the variations were in contiguous genomic loci". Can authors also provide a little bit discussion for this ?

Rome, 14th January 2022

Dear Editor,

Enclosed please find the revised version of manuscript “**Genome analysis of *Legionella pneumophila* ST23 from various countries reveals highly similar strains**” (Ref. N.: **LSA-2021-01117-T**)

Here are the point-by-point responses to reviewers' comments.

Reviewer #1:

The manuscript submitted by Ricci et al. examines the genomic sequences of several dozens of the bacteria '*Legionella pneumophila*' serogroup 1 (Lp1) sequence type (ST) 23, obtained from clinical and environmental isolates, and responsible for epidemic and sporadic cases of Legionary Disease between 1995 and 2018 in Italy. The authors observe that all of the isolates are closely related, with little variation, mainly due to recombination events. The authors then conclude a clonal origin of the italian ST23 type.

I think this work has merit. It produces 62 new genomic sequences of a pathogen of interest in Italy, Europe and also worldwide. The general approach of genomic epidemiology is proving to be very fruitful as it develops very swiftly. Nevertheless, I have some major concerns about the extent of the validity of the general conclusions of this manuscript. My doubts are mainly driven by issues regarding taxon sampling and phylogenetic analyses. I will explain myself.

First, concerning the taxon sampling issue, more genomic sequences of Lp1 ST23, and probably ST2695 and other STs, should be included in the analyses. Indeed, the variability observed within the italian genome sequences is as high as the observed between some italian and the other european sequences examined in this study (Sup. Table 1). Hence, a more thorough taxon sampling should be carried out in order to adequately assess variability, transmissions and origins of the italian variants. These data can be obtained from studies as David et al., 2017(<https://doi.org/10.1371/journal.pgen.1006855>) or crossing databases as PATRIC <https://www.patricbrc.org/>) and LegionellaDB (<https://doi.org/10.1016/j.tim.2021.01.015>).

Answer:

Thank you very much for your comments and suggestions.

We agreed and extended the analyses to further 49 ST23, including eight EUL strains and 14 not-ST23. We did not have further ST2695. We have also done the cgMLST and SNP analyses that include these new isolates.

In regards to the phylogenetic analyses, the phylogenetic tree from figure 4 is not appropriately presented. First, the tree inference method is not described. Second, no EUL000* data was presented. A new tree rooted and inferred correctly should be done including the newly selected genomes and a potential outgroup. In any case, David et al (2017) mentioned above show a clear methodology on how to analysis SNP data. In addition, the color code of geographic regions should be maintained in the wgMLST and also in the SNP analyses. It could be helpful to see the year of isolation, which could be included in the tip labels.

Answer:

Figure 4 was re-produced and now is Figure S1. The tree inference method is described in Materials and Methods section and EUL* have been also indicated. It has been more difficult to maintain the color code of geographic regions, because it has produced during the online phyloviz analysis. However, we have tried to evidence the most important region by circles, the year of isolation was not added because the Figure resulted too much confusing.

With these new results at hand, it would be important to compare wgMLST and SNP phylogenetic approaches and discuss the eventual differences between them.

Answer:

Yes, cgMLST and SNP were compared and discussed.

Finally I think that, given the amount of data generated, deeper analyses could be carried out. In addition to the anecdotal description of the differences in alleles some genomic comparisons could be done so as to compare gains and losses of the isolates of interest, as well as differences in gene complement which can be connected to virulence or survival. To perform this analyses inhouse could be somewhat demanding of bioinformatic resources; however, reasonable and meaningful results of "blast atlases" and "core" and "accessory" genomes can be obtained with online servers such as GView (<https://server.gview.ca/>).

Answer:

Thanks for this suggestion. We could just determine the pangenome but we were not able to perform further analyses.

There are many typos in the manuscript and I think it could benefit from an editing process.

Answer:

An editing was done.

Summarizing, in my opinion the data presented in this work is valuable but needs a more thorough analysis to clarify certain claims of the article, and also to complement the descriptive work of the observed natural variation of this pathogen connected to clinical and environmental conditions in epidemic and sporadic contexts.

Reviewer #2:

This work sought to analyze the *Legionella pneumophila* serogroup 1 (Lp1) sequence type (ST) 23 at genomic level. A core genome multi locus sequence typing (cgMLST), based on a previously described set of 1521 core genes, and single nucleotide polymorphisms (SNPs) approaches were applied to the Lp1 ST23 strain collection. Although very few loci and single nucleotide variations occurring in ST23 genomes, they found that most of the variations were located in adjacent genomic regions due to recombination events. The evidence is sound, and the conclusion is well supported by the data presented in the work

I only have several minor comments for discussion of this work:

1. It is not surprised to see that very few variations were found in ST23 genomes. However, it would be interesting to investigate or, at least, to discuss whether there are any genotype hot spot(s) related with the drug-resistance of LP?

Answer:

Thank you for your comment.

We did further experiments including more ST23 and not-ST23 isolates. We did not perform analysis of hot spots but we could observe that the core gene differences involved especially genes encoding metabolic enzymes, binding or transport proteins.

2. They found that "the 99.7% of SNP had been acquired by recombination". Can authors discuss a little bit more why LP acquire most of SNPs via recombination?

Answer:

The discussion has been completely reworked even though on the basis of the investigation done it was difficult to discuss why Lp acquire SNP by recombination.

3. They found that "most of the variations were in contiguous genomic loci". Can authors also provide a little bit discussion for this?

Answer:

The recombination events typically involve contiguous genomic loci because they are the result of genetic exchanges with transfer of blocks of genetic material.

February 8, 2022

RE: Life Science Alliance Manuscript #LSA-2021-01117-TR

Dr. Maria Scaturro
Istituto Superiore di Sanità
Viale Regina Elena, 299
Rome 00161
Italy

Dear Dr. Scaturro,

Thank you for submitting your revised manuscript entitled "Genome analysis of Legionella pneumophila ST23 from various countries reveals highly similar strains". We would be happy to publish your paper in Life Science Alliance pending final revisions necessary to meet our formatting guidelines.

- please upload your Tables in editable .doc or excel format;
- please add the Twitter handle of your host institute/organization as well as your own or/and one of the authors in our system
- main tables should be included at the bottom of the main manuscript file or be sent as separate files.
- please add your main, supplementary figure, and table legends to the main manuscript text after the references section
- please consult our manuscript preparation guidelines <https://www.life-science-alliance.org/manuscript-prep> and make sure your manuscript sections are in the correct order and labeled correctly
- please add an Author Contributions section to your main manuscript text
- there is a callout for figure S2, although the figure is not provided, while call out for figure S1 is missing from the manuscript text. Please revise
- please add callouts for Figure 6A-D to your main manuscript text
- conflict of interest is present but not headed. Please do it after the authors contribution section

FIGURE CHECKS:

- the quality of figure S1 should be increased

A. FINAL FILES:

B. MANUSCRIPT ORGANIZATION AND FORMATTING:

Sincerely,

Reviewer #1 (Comments to the Authors (Required)):

After carefully reading the PBP response and the new version of the manuscript I find that most of the important points raised in my previous review have been addressed, hence, to my opinion, the manuscript met the standards to be published.

Reviewer #2 (Comments to the Authors (Required)):

Authors have addressed my previous comments. I have no more comment.

February 15, 2022

RE: Life Science Alliance Manuscript #LSA-2021-01117-TRR

Dr. Maria Scaturro
Istituto Superiore di Sanità
Viale Regina Elena, 299
Rome 00161
Italy

Dear Dr. Scaturro,

Thank you for submitting your Research Article entitled "Genome analysis of Legionella pneumophila ST23 from various countries reveals highly similar strains". It is a pleasure to let you know that your manuscript is now accepted for publication in Life Science Alliance. Congratulations on this interesting work.

DISTRIBUTION OF MATERIALS:

Again, congratulations on a very nice paper. I hope you found the review process to be constructive and are pleased with how the manuscript was handled editorially. We look forward to future exciting submissions from your lab.

Sincerely,
